
# An assessment of macrophysical and microphysical cloud properties driving radiative forcing of shallow trade-wind clouds

Anna E. Luebke[1], André Ehrlich[1], Michael Schäfer[1], Kevin Wolf[1,2], and Manfred Wendisch[1]

[1]Leipzig Institute for Meteorology, University of Leipzig, Leipzig, Germany
[2]Laboratory for Atmospheric and Space Physics, University of Colorado, Boulder, Colorado, USA

**Correspondence:** Anna E. Luebke (anna.luebke@uni-leipzig.de)

**Abstract.** The clouds in the Atlantic trade-wind region are known to have an important role in the global climate system. Acquiring a comprehensive characterization of these clouds based on observations is a challenge, but it is a necessary piece of information for the evaluation of their representation in models. An exploration of how the macrophysical and microphysical cloud properties and organization of the cloud field impact the large-scale cloud radiative forcing is presented here. Direct

measurements of the cloud radiative effects from the Broadband AirCrAft RaDiometer Instrumentation (BACARDI) on board the High Altitude and LOng Range Research Aircraft (HALO) and cloud observations from the GOES-16 satellite during the Elucidating the role of clouds-circulation coupling in climate (EUREC$^4$A) campaign provide evidence to demonstrate what drives the cloud radiative effects in shallow trade-wind clouds. We find that the solar and terrestrial radiative effects of these clouds are largely driven by their macrophysical properties (cloud fraction and a scene-averaged liquid water path). However,

we also conclude that the microphysical properties, cloud top height and the organization of the cloud field demonstrate an increasing relevance in determining the cloud radiative effects as the cloud fraction increases.

## 1   Introduction

Shallow, marine trade-wind cumuli in the tropics have been established as a key component in influencing the radiative energy budget of the Earth's atmosphere in response to a changing climate (Bony et al., 2017; Klein et al., 2017; Stevens et al., 2021).

Through various research efforts, the microphysical, macrophysical and radiative properties of these clouds have been mapped out, for example with the aid of high resolution satellite observations (Zhao and Di Girolamo, 2007; Mieslinger et al., 2019) and through dedicated research campaigns such as the Cloud, Aerosol, Radiation and tuRbulence in the trade regime over Barbados (CARRIBA; Siebert et al., 2013) and the Next-Generation Aircraft Remote Sensing for Validation field studies (NARVAL I and NARVAL II; Stevens et al., 2019). Such observational data sets are ideal for the purposes of process studies and the validation

and improvement of numerical weather prediction models (NWPs) and global climate models (GCMs). However, constraining a comprehensive characterization of shallow trade-wind clouds based on their macrophysical and microphysical properties and assessing the associated dynamics remains difficult. An often cited reason for this is the lack of representative observations (e.g. Zhang et al., 2013; Herwehe et al., 2014; Khain et al., 2015; Bony et al., 2017). Specifically, though model studies are able to quantify the radiative budget, it is challenging to obtain the suitable microphysical and radiative measurements for



verification. Thus, the impact of shallow trade-wind clouds on the atmospheric radiative energy budget in a warming world remains a critical topic.

    Despite this challenge, the influence of cloud properties on their radiative effects is already well known. For solar (shortwave) radiative effects, cloud fraction has been established as a useful parameter for estimating the radiative properties or effects of the clouds for a given scene (e.g., Chen et al., 2000), particularly by using the relation between albedo, which is the combination

of cloudy- and cloud-free-sky albedo, and cloud fraction (George and Wood, 2010; Bender et al., 2011, 2016; Engström et al., 2015). However, this relation can be further influenced by variations in liquid water path (LWP) when cloud fraction is held constant (Bender et al., 2016). Cloud microphysical characteristics have been found to have less of an impact, but George and Wood (2010) suggest that these impacts may be masked by the complex interactions between macrophysical and meterological variability. Assessments of the cloud albedo demonstrate how changes in aerosol concentrations (Peng et al., 2002; Diamond

et al., 2020), and thus droplet size via the Twomey effect (Twomey, 1977; Werner et al., 2014), enhances cloud albedo and cloud optical thickness due to a decrease in droplet effective radius ($r_{eff}$). However, this relation between $r_{eff}$ and cloud albedo differs between optically thick and thin clouds (Lohmann et al., 2000), likely stemming from a dependence on whether the cloud is precipitating, which impacts LWP and the droplet size distribution. In optically thick clouds, cloud albedo increases with decreasing $r_{eff}$, while cloud albedo decreases with decreasing $r_{eff}$ in optically thin clouds. The spatial distribution or

organization of the cloud field, which is strongly related to cloud fraction and LWP, also impacts the resulting albedo (Wood and Hartmann, 2006; McCoy et al., 2017).

    With regard to terrestrial (longwave) cloud radiative effects, macrophysical properties are known to have the greatest influence. Cloud top height (or temperature) has been found to have the most impact, but cloud amount is also important (Ardanuy et al., 1988; Chen et al., 2000). As described by Chen et al. (2000), because cloud top height is so critical to terrestrial cloud

radiative effects, the cloud type is also known be an important factor. Among shallow clouds, stratocumulus along with stratus and altostratus, have the greatest impact on surface terrestrial effects. The organization of the cloud field also plays a role. For example, Tobin et al. (2012) found that the outgoing terrestrial radiation of deep convective clouds decreased as the cloud field became more aggregated. Whether this holds true for shallow trade-wind clouds is explored in this study.

    Thus far, the observational research in this area has largely relied on satellite-based measurements of the cloud radiative

effects, which has the benefit of providing long-term, global observations, but often with the trade-off of a coarser spatial and temporal resolution. Furthermore, the radiative flux densities are usually derived from combinations of radiance observations and radiative transfer models or the Clouds and the Earth's Radiant Energy System (CERES) instruments. This means that irradiance is not measured directly, which is one reason why airborne observations of this quantity are so important. The recent ElUcidating the RolE of Cloud-Circulation Coupling in ClimAte (EUREC[4]A) campaign in 2020 used multiple platforms (e.g.

airborne and satellite) and instrumentation methods to assess the cloud field from multiple angles, both literally and with regard to research topics, and with a strong degree of collocation (Stevens et al., 2021). Among these measurements are airborne observations of solar and terrestrial broadband irradiance.

    With these new radiation observations, the major objective of this study is to understand how and to what degree the macro-physical and microphysical cloud properties influence the cloud radiative effects of shallow trade-wind clouds. Furthermore,





we seek to understand their relative importance in driving the cloud radiative effects and how different properties might work together or against each other to result in a specific cloud radiative effect. In Sect. 2, the relevant instrumentation, data products, and analysis methods for this study are discussed. The results of the analysis are presented in Sect. 3 with different approaches to exploring the data and the relations between the cloud properties and the solar and terrestrial cloud radiative effects. Section 4 concludes by summarizing and discussing the results and their implications as well as providing the context for future work.

## 2  Methodology

The analysis presented here is based on data taken during the EUREC$^4$A campaign (Bony et al., 2017; Stevens et al., 2021), which took place in the winter of 2020. During this season, the trade-wind region is characterized by more cloud cover and stratiform structures in comparison to the rather convective, wet summer season (Nuijens et al., 2014). The general aim of this campaign was to sample the clouds and large-scale dynamics in the trade-wind region of the Atlantic Ocean, just off the coast of Barbados. With the participation of four research vessels, five aircraft, three remote sensing stations on Barbados and many unmanned research crafts, the synergistic approach of this campaign offered the opportunity to observe the trade-wind region from several perspectives. Specifically, data from the High Altitude and LOng Range Research Aircraft (HALO) operated by the DLR (Deutsches Zentrum für Luft- und Raumfahrt, e.V.) and the Geostationary Operational Environmental Satellite 16 (GOES-16) operated by the National Oceanic and Atmospheric Administration (NOAA) are utilized in this study.

### 2.1  Airborne observations using HALO

As in NARVAL I and NARVAL II, which previously took place in this region, an instrument payload known as the cloud-observatory configuration was installed on HALO (Stevens et al., 2019). This payload consists primarily of remote sensing instruments, comparable to the instrumentation of the current A-Train satellite constellation and the upcoming Earth Cloud Aerosol and Radiation Explorer (EarthCARE) satellite. For EUREC$^4$A, the instrumentation was extended with the Broadband AirCrAft RaDiometer Instrumentation (BACARDI; Zöger et al., 2021) and the Video airbornE Longwave Observations within siX channels (VELOX; Schäfer et al., 2021) system. This instrument payload was designed to observe the cloud population as well as the surrounding environment (i.e. dynamic and thermodynamic conditions) simultaneously. Active and passive sensors provide a characterization of the cloud properties – microphyiscal, macrophysical and radiative – in vertical and horizontal dimensions, while methodically released dropsondes along the flight path obtain in situ measurements of the vertical atmospheric profile. See Konow et al. (2021) for further information concerning the instrumentation and flights of HALO during EUREC$^4$A.

For the purposes of the campaign, specifically as a strategy for properly capturing the scale of the dynamics influencing cloud development, HALO primarily flew in a pattern defined by clockwise circles repeated in the same location, approximately 220 km in diameter and requiring an hour to complete. This flight strategy was independent of the meteorological situation. Other flight patterns were also observed for additional purposes, but the work here focuses on the circles so as to capture and constrain the observations in a single location.





The radiative component of this analysis is estimated from observations from BACARDI (Konow et al., 2021). This new radiometer package comprises two sets of Kipp and Zonen pyranometers (CMP22) and pyrgeometers (CGR4) mounted to the aircraft fuselage. Together, they provide measurements of the upward and downward solar (sensitive to wavelengths between 0.2 –3.6 µm) and terrestrial (4.5 –42 µm) broadband irradiance at flight level. The respective uncertainty of the measurements from these sensors is 1 % and 4 %. The data used here has been corrected in post-processing for temperature dependence and response time of the sensors as well as aircraft attitude, similar to the methods described by Ehrlich and Wendisch (2015).

## 2.2 Satellite observations from GOES-16

The cloud properties in this analysis come from the data products of the GOES-16 geostationary satellite Advanced Baseline Imager (ABI) (Schmit et al., 2017, 2018). GOES-16 is part of the GOES-R series of satellites. Throughout the duration of the campaign, GOES-16 was operated to produce images of the EUREC$^4$A research domain at a temporal resolution of 1 minute, when possible, which turns out to be a strong benefit for analyzing the evolution of the clouds. Specifically, we utilize the cloud mask product (Heidinger and Straka, 2013) as well as retrieved LWP, $r_{\text{eff}}$ (Walther et al., 2013) and cloud top height ($z_{\text{ct}}$; Heidinger, 2012) products. The spatial resolution for the data products used here is 2x2 km at nadir. The cloud mask shows good agreement with the Cloud-Aerosol Lidar and Infrared Pathfinder Satellite Observation (CALIPSO) over the Contiguous United States, but low clouds (cloud tops below 2 km) were the most commonly missed cloud type (Jiménez, 2020). Also, further evaluation shows that the increased pixel resolution relative to its predecessor, GOES-13, results in an improved $r_{\text{eff}}$ retrieval in comparisons with airborne measurements (Painemal et al., 2021). It is therefore expected that the LWP retrieval is also improved, however, both products continue to have a high bias.

For the application of the cloud mask product from GOES-16, pixels labeled as "probably cloudy" and "cloudy" are accepted as cloud pixels in the calculation of the cloud fraction. The cloud mask is also used to calculate the organization of the cloud field, which is described by the organization index, $I_{\text{org}}$ (Weger et al., 1992; Tompkins and Semie, 2017). This index classifies a given cloud scene as regular, random, or clustered. The classification is based on statistics of the nearest neighbor (NN) distances between clouds in comparison to a theoretical scene with a random distribution, which can be described as a Poisson point process. If the NN distances are on average larger than they would be if the clouds were randomly distributed ($I_{\text{org}} < 0.5$), then the scene is regular. If they are smaller ($I_{\text{org}} > 0.5$), the scene is classified as clustered. An $I_{\text{org}}$ value of 0.5 indicates that the scene is random. More detailed explanations can be found in Tompkins and Semie (2017) and Mieslinger et al. (2019).

## 2.3 Deriving cloud radiative effects

Cloud radiative forcing ($\Delta F$) is defined as the difference between the observed net (downward - upward) irradiance ($F^{\downarrow}$ - $F^{\uparrow}$) at an altitude ($z$, in this case flight altitude) and the net irrandiance in cloud-free conditions, as shown in Eq. 1 (Stapf et al., 2021):

$$\Delta F(z) = \left[F^{\downarrow}(z) - F^{\uparrow}(z)\right]_{\text{cloud}} - \left[F^{\downarrow}(z) - F^{\uparrow}(z)\right]_{\text{cloud-free}}. \tag{1}$$





$\Delta F$ can also be further divided into its terrestrial ($\Delta F_{\mathrm{ter}}$) and solar ($\Delta F_{\mathrm{sol}}$) components, which are defined in Eqs. 2 and 3, respectively:

$$\Delta F_{\mathrm{ter}}(z) = \left[ F_{\mathrm{ter}}^{\downarrow}(z) - F_{\mathrm{ter}}^{\uparrow}(z) \right]_{\mathrm{cloud}} - \left[ F_{\mathrm{ter}}^{\downarrow}(z) - F_{\mathrm{ter}}^{\uparrow}(z) \right]_{\mathrm{cloud-free}}, \tag{2}$$

$$\Delta F_{\mathrm{sol}}(z) = \left[ F_{\mathrm{sol}}^{\downarrow}(z) - F_{\mathrm{sol}}^{\uparrow}(z) \right]_{\mathrm{cloud}} - \left[ F_{\mathrm{sol}}^{\downarrow}(z) - F_{\mathrm{sol}}^{\uparrow}(z) \right]_{\mathrm{cloud-free}}. \tag{3}$$

For $\Delta F_{\mathrm{sol}}$, Eq. 3 can be further simplified with the assumption that $F_{\mathrm{sol}}^{\downarrow}$ is identical in cloudy and cloud-free cases while flying above cloud:

$$\Delta F_{\mathrm{sol}}(z) = - \left[ F_{\mathrm{sol,cloud}}^{\uparrow}(z) \right] - \left[ -F_{\mathrm{sol,cloud-free}}^{\uparrow}(z) \right]. \tag{4}$$

However, $\Delta F_{\mathrm{sol}}$, is a function of solar zenith angle (SZA). For this reason, albedo (Eq. 5) is a more useful parameter to describe solar radiative fluxes because it expresses the relative difference between $F_{\mathrm{sol}}^{\downarrow}$ and $F_{\mathrm{sol}}^{\uparrow}$, thereby avoiding the SZA dependence:

$$\alpha = \frac{F_{\mathrm{sol}}^{\uparrow}}{F_{\mathrm{sol}}^{\downarrow}}. \tag{5}$$

With this in mind, the dependence of $\Delta F_{\mathrm{sol}}$ on SZA can also be addressed. The same strategy applied to Eq. 3, wherein the cloud-free fluxes are subtracted from the total to isolate the cloud effect, can also be used with albedo (Eq. 5) to isolate the cloud effect on the observed albedo ($\alpha_{\mathrm{ce}}$). Simply, the cloudy and cloud-free $F_{\mathrm{sol}}^{\uparrow}$ in Eq. 4 are normalized with $F_{\mathrm{sol}}^{\downarrow}$, which removes the SZA dependence, thus making observations from different SZAs comparable:

$$\alpha_{\mathrm{ce}}(z) = - \left[ \frac{F_{\mathrm{sol,cloud}}^{\uparrow}(z)}{F_{\mathrm{sol}}^{\downarrow}(z)} \right] - \left[ -\frac{F_{\mathrm{sol,cloud-free}}^{\uparrow}(z)}{F_{\mathrm{sol}}^{\downarrow}(z)} \right]. \tag{6}$$

Since measurements in cloud-free conditions cannot be obtained simultaneously or in similar conditions to those observed during each flight, simulated cloud-free cases of each flight are calculated using the libRadtran software package (Emde et al., 2016). Dropsondes along the flight track provide measured profiles of thermodynamic atmospheric properties (humidity, temperature) used in the simulations (George et al., 2021). Estimations of the uncertainties of these simulations are given in Table 1. The largest source of uncertainty is found to be the atmospheric data, particularly for the cloud-free $F_{\mathrm{ter}}^{\uparrow}$. Due to the limited spatial resolution of the dropsondes, this method cannot always replicate an observed cloud-free situation. Additionally, because the sondes may pass through clouds on their way down, the humidity and temperature profiles used for the simulation could be biased. In order to minimize these effects, only the simulations at the dropsonde locations are used to approximate a cloud-free $F_{\mathrm{ter}}^{\uparrow}$, which is then interpolated for the rest of the flight track.

## 2.4 Combining satellite and aircraft observations

In the case of the analysis presented here, only a subset of seven flights are used, which are described in Table 2. Those times when data from both BACARDI and GOES-16 were available are given there. Unfortunately, this also means that only



**Table 1.** Estimations of the uncertainty of the simulated cloud-free conditions based on different possible sources.

| Source of Uncertainty | Wavelength Range | Parameter | Uncertainty |
|---|---|---|---|
| Atmospheric Profile (dropsonde) | Solar | $F^{\downarrow}$ | < 1 % |
| | | $F^{\uparrow}$ | 3 % |
| | Terrestrial | $F^{\downarrow}$ | 6 % |
| | | $F^{\uparrow}$ | 8 % |
| Surface Temperature ($\pm 3\,\mathrm{K}$) | | $F^{\downarrow}$ | < 1 % |
| | | $F^{\uparrow}$ | 2.5 % |

**Table 2.** The flights from EUREC[4]A that are used in this analysis are given here with their respective flight IDs (HALO-mmdd) along with the times when HALO was in the circling flight pattern and GOES-16 data were available. The amounts of time and number of GOES images are for cloudy and cloud-free sky, but a total amount of time and number of GOES images for cloudy sky only is given as well. The general descriptions of the flights come from notes in the flight reports.

| Flight ID | Time (UTC) | Duration (hh:mm) | GOES Images | General Description |
|---|---|---|---|---|
| HALO-0124 | 11:52–16:56 | 5:05 | 305 | Shallow Cu, Sc, deeper convection |
| HALO-0128 | 15:18–18:38; 19:53–20:21 | 3:40 | 220 | Shallow Cu clusters and lines |
| HALO-0131 | 15:28–18:50; 19:56–20:23 | 3:51 | 231 | Dust, suppressed shallow Cu, "fish" clusters, Sc |
| HALO-0202 | 18:29–19:34 | 1:06 | 66 | Large "flowers", some dissipating |
| HALO-0207 | 15:00–15:34; 17:14–20:23 | 3:45 | 225 | Shallow Cu, "flowers", precipitation |
| HALO-0209 | 11:47–12:56; 13:54–17:24 | 4:41 | 281 | Shallow and deeper Cu, cold pools, precipitation |
| HALO-0211 | 12:57–18:55 | 5:58 | 358 | Convection with precipitation, Sc, shallow Cu |
| **Total** | | **28:06** | **1686** | |
| **Total (cloudy scenes only)** | | **14:02** | **842** | |

partial flights are usable in some cases. Additional filters were applied regarding the SZA and the presence of clouds above the aircraft (e.g. cirrus). To prevent complications in the case of low sun conditions and to avoid the need to consider cirrus in this study, the data were limited to SZAs less than 70° and to flights with cloud-free conditions above the flight altitude of HALO. Furthermore, the maximum mean $z_{\mathrm{ct}}$ for an observed scene is limited to 4 km to avoid the inclusion of deeper convective clouds, which have properties and associated relationships that should be considered separately. Scenes without clouds below HALO are also excluded.

To enable the combined use of BACARDI and GOES-16 data, spatial collocation needs to be assured with both observations covering the same footprint at the same time. BACARDI has a 180° field of view (FOV), but radiation from different directions is weighted with the viewing zenith angle. Based on geometry, 95 % of the upward irradiance is determined by radiances




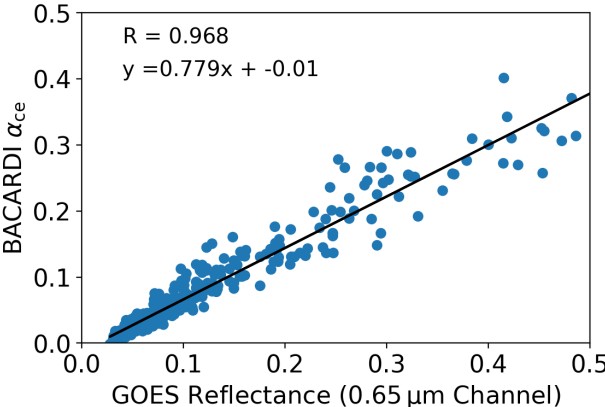

**Figure 1.** Comparison of $\alpha_{ce}$ from BACARDI and the mean reflectance from the 0.65 µm channel of GOES-16 for a FOV of 51° from the flight on 24 January 2020.

160 within a FOV of approximately 77°. However, such a footprint may not necessarily be compatible with the radiation measured by GOES-16. Therefore, as a quality check, we compared the calculated $\alpha_{ce}$ from BACARDI with the mean reflectance from the 0.65 µm channel for footprints of varying sizes cut out from GOES-16. Because we are comparing different quantities, they will not match exactly, but we assume that the footprint size at which they show the best correlation represents a footprint size where GOES-16 observations are comparable with BACARDI data. A FOV of 51° was found to provide the best correlation

165 (R = 0.968). An example of this comparison from the flight on 24 January 2020 for a FOV of 51° is shown in Fig. 1. Based on the selected FOV and a typical flight altitude of 10 km, the most appropriate footprint size has a radius of approximately 13 km.

Additionally, the cloud scenes observed over the course of each minute are oblong instead of perfectly circular due to the movement of the aircraft. Over the course of a minute, the aircraft travels approximately 12 km with an average ground speed

170 of 200 m s$^{-1}$, so the area of each one minute scene is approximately 820 km$^2$. Examples of the GOES-16 cloud mask product and the corresponding footprint cutout from the flight on 24 January 2020 are shown in Fig. 2.

Mean values are used to describe the cloud properties of each scene. The individual quantities are weighted with the cosine of the viewing angle across the footprint of BACARDI before calculating a mean value. This strategy is used to capture an expression of the properties that is compatible with the radiative view that BACARDI has (i.e. clouds on the periphery make

175 less of an impact than clouds in the nadir view of the instrument). Furthermore, two representations of the LWP are used – mean LWP calculated for the cloud pixels only (LWP$_{cloud}$) and a mean LWP calculation that includes both the cloudy and cloud-free pixels within the footprint-sized cloud scene (LWP$_{scene}$). LWP$_{scene}$ considers LWP in a more macrophysical sense because of its relationship to cloud fraction and the fact that it represents the amount of water distributed over the scene, while LWP$_{cloud}$ considers LWP as microphysical in nature.





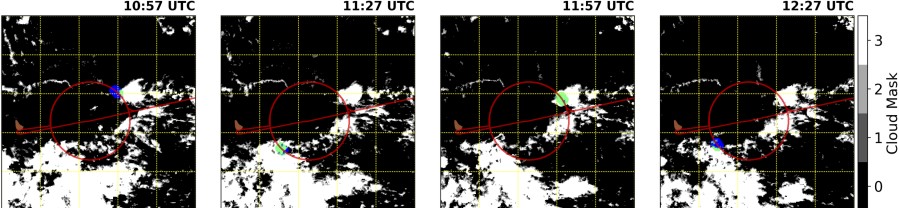

**Figure 2.** Four example scenes from the flight on 24 January 2020. The cloud mask product from GOES is displayed, with the blue (cloud free) and green (cloud) colors indicating the collocated 51° footprint of BACARDI. The solid red line indicates the flight track. The island of Barbados is shown in brown for reference.

The size of the footprint cut out from the full GOES scene is not sufficient for a meaningful quantification of the cloud organization. Trade-wind cumuli typically have sizes (measured as maximum diameter) from a few hundred meters to less than 2 km (Schnitt et al., 2017), so only a few clouds are present within the GOES cutouts. However, assuming that the cloud organization is driven on larger scales, we used a larger domain size of 5°x5° for this assessment of organization. Given the coarse resolution of GOES-16 this consideration seems appropriate. The full example scenes in Fig. 2 are the larger domain size, but for the analysis, the center of each larger domain would be aligned with the location of HALO at that time.

## 3 Sensitivity of cloud radiative effects to cloud properties

### 3.1 Overview of cloud data

The radiative data in this study was recorded by the BACARDI instrument and the cloud properties were obtained from GOES-16 satellite data products (see Sects. 2.1 and 2.2). A set of histograms depicting the distributions of the data is given in Fig. 3. The mean $\alpha_{ce}$ is 0.086, while the mean $\Delta F_{ter}$ is 6.35 Wm$^{-1}$. The mean $\Delta F_{sol}$ is -77.64 Wm$^{-1}$, which demonstrates that the solar part of the cloud radiative forcing of shallow trade-wind cumuli is significantly stronger than the forcing from the terrestrial part. However, due to the influence of SZA on $\Delta F_{sol}$, the mean value is provided only as an estimate for the purposes of comparison. The mean cosine weighted cloud fraction is 0.204, which is larger than the value of 0.087 reported by Mieslinger et al. (2019) from Advanced Spaceborne Thermal Emission and Reflection Radiomater (ASTER) imagery of trade-wind cumuli. Differences are likely to stem from the exclusion of non-cloudy scenes in this study as well as the small footprint and coarseness of the GOES satellite resolution. The mean $z_{ct}$ of 1.9 km found here is slightly higher than the 1.3 km reported in their study, and the cloud field is found to be less clustered on average (I$_{org}$ of 0.74 versus 0.89). The overall observed values are within similar ranges, so the smaller sample size within a single season may also be the cause of differences to previous studies. The mean LWP$_{scene}$ and mean LWP$_{cloud}$ are 19 gm$^{-2}$ and 98 gm$^{-2}$, respectively. For comparison, during the NARVAL campaigns, retrievals of LWP from remote sensing instrumentation reported a mean LWP from the sampled clouds of about 63 gm$^{-2}$ during the dry, winter season (Jacob et al., 2019). The fact that thicker clouds (LWP > 50 gm$^{-2}$) were also more





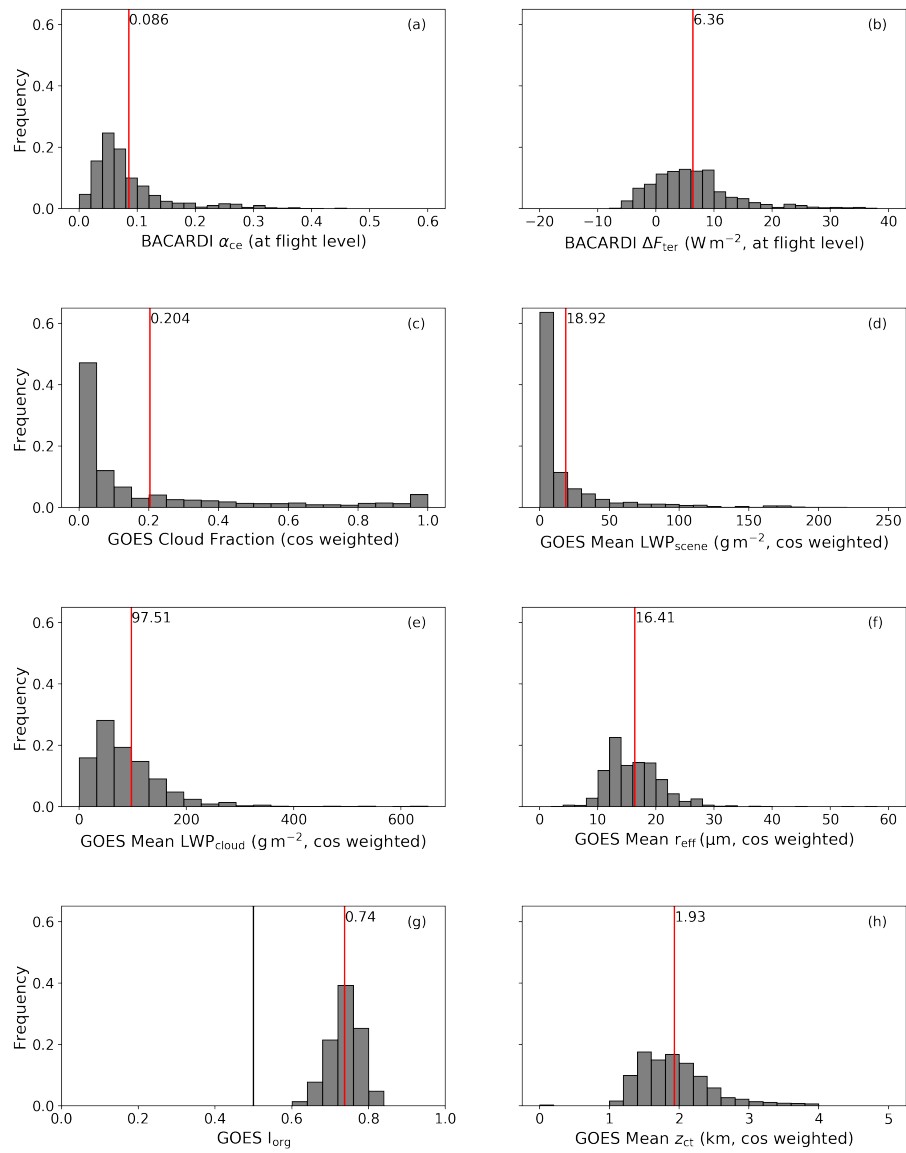

**Figure 3.** Histograms of all cloud properties for the subset of EUREC[4]A flights and data used in the analysis. Labeled red bars indicates the mean value. (a) Effect of clouds on albedo (b) Terrestrial cloud radiative forcing (c) Cloud fraction (d) Mean liquid water path for the scene (e) Mean liquid water path for the clouds (f) Mean effective radius (g) Organization index; black bar at 0.5 separates normal and clustered $I_{org}$ values (h) Mean cloud top height.

frequent during this season was also noted. Mean $r_{eff}$ in this study is 16 µm, which is comparable to previous observations in the vicinity of Barbados (10–17 µm in pristine cases; Werner et al., 2014).





## 3.2 Observed relations between cloud properties and cloud radiative effects

The impact of cloud properties on the radiation budget was analyzed separately for the solar and terrestrial components of the observed radiation because these two components are known to interact differently with cloud properties. A series of scatter plots in Figs. 4 and 5 demonstrate the relationships of $\alpha_{ce}$ and $\Delta F_{ter}$, respectively, to the various cloud properties analyzed here.

In Fig. 4, it becomes apparent that the macrophysical properties (cloud fraction and $LWP_{scene}$) demonstrate the most obvious
relationship to $\alpha_{ce}$ – as $\alpha_{ce}$ increases, the cloud fraction and $LWP_{scene}$, panel (a) and (b), also increase, as expected. The relation between $\alpha_{ce}$ and $z_{ct}$ (Fig. 4(f)) is more weakly correlated, but generally as $z_{ct}$ increases, so does $\alpha_{ce}$. In the case of $LWP_{cloud}$ (Fig. 4(c)), two populations emerge – one where LWP and $\alpha_{ce}$ are strongly positively correlated and one where an increase in LWP does not appear to impact $\alpha_{ce}$. This second branch may represent cases where changes in microphysical properties are less relevant to the radiative impact of the clouds. From the perspective of a scatter plot, the relationships of $\alpha_{ce}$ to $r_{eff}$ and $I_{org}$
are unclear given the very weak correlations (Fig. 4(d) and Fig. 4(e)).

Based on these results, we would expect to see the same patterns to be observed for individual flights as well. The different colored points and bars in Fig. 4 depict the median values (see also Table 3) of each parameter for each of the seven investigated flights and the range from the 10th to 90th percentiles of the different distributions, respectively. The clouds observed during the flight on 28 January 2020 (red star) have the lowest $\alpha_{ce}$, while the highest median $\alpha_{ce}$ was observed during the flight on
24 January (black circle). The expectation that follows is that the cloud properties will also display a pattern according to their relationship to $\alpha_{ce}$. For example, properties like cloud fraction and $LWP_{scene}$ are also generally increasing in a similar flight order as the $\alpha_{ce}$, with 28 and 31 January showing the some of the lowest cloud fraction and $LWP_{scene}$ values and 24 January showing the highest. However, the observations on 7 February (green plus) do not follow this trend for both cloud fraction and LWP. The remaining properties ($LWP_{cloud}$, $r_{eff}$, $I_{org}$ and $z_{ct}$) appear not to follow any pattern given the order of the flights with
increasing $\alpha_{ce}$. This demonstrates again that the macrophysical properties are more closely linked to the observed $\alpha_{ce}$ than the microphysical properties.

From this series of panels in Fig. 4, it is also possible to pick out details about each flight and identify relationships between cloud properties and $\alpha_{ce}$. For example, the $\alpha_{ce}$ values measured during the flight on 31 January (orange triangle) fall on the relatively lower end of observed $\alpha_{ce}$ as well as similarly low cloud fraction and $LWP_{scene}$. The mean $LWP_{cloud}$ is relatively
high and the $r_{eff}$ relatively low. On this day, the cloud field was characterized as quite clustered with lower cloud tops. During this flight, the cloud field was populated with suppressed shallow cumulus, so-called "fish" structures (Schulz et al., 2021) and lots of dust. The dust could be an explanation for why the $r_{eff}$ is quite small, and the notion of suppressed shallow cumulus fits to the lower cloud top heights. While small droplets typically indicate higher $\alpha_{ce}$, the characteristics of the macrophysical properties seem to be more dominant in this case. In contrast, the flight on 7 February (green plus) has a lower $\alpha_{ce}$ relative to
other flights, but has a higher cloud fraction and $LWP_{scene}$. The $LWP_{cloud}$ is also relatively lower, the $r_{eff}$ is relatively larger, the cloud tops are higher and the cloud field is less clustered than most flights. This flight was dominated by cloud structures known as "flowers" (Schulz et al., 2021) and precipitation was noted. Thus, it could be possible that despite the higher cloud fraction



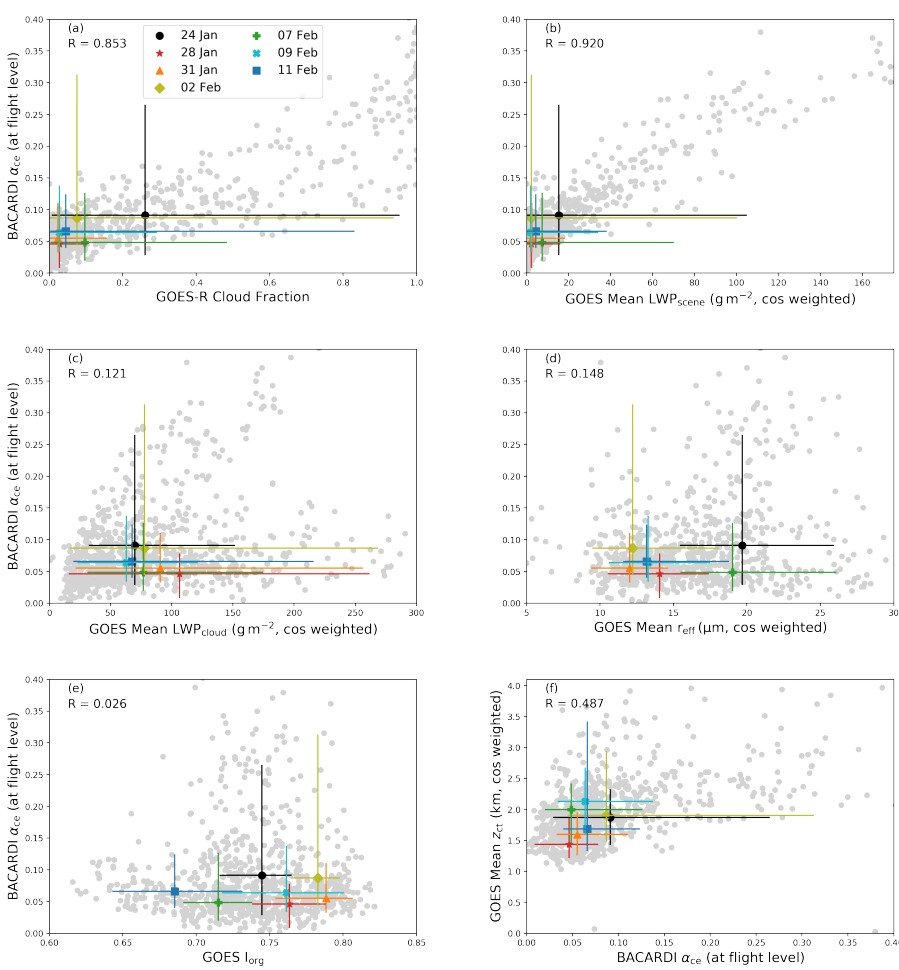

**Figure 4.** (a) - (f) Scatter plots (gray) of different cloud properties as a function of $\alpha_{ce}$ for the subset of EUREC$^4$A flights used in the analysis. Different colored shapes denote the median observed values from individual flights, while the associated bars indicate the range of values between the 10th and 90th percentiles. Plots are zoomed in to include at least the 10th–90th percentile ranges for each variable.





**Table 3.** Median values shown in Figs.4 and 5. The highest and lowest values for each parameter are marked in bold.

|  | 24 Jan | 28 Jan | 31 Jan | 02 Feb | 07 Feb | 09 Feb | 11 Feb |
|---|---|---|---|---|---|---|---|
| $\alpha_{ce}$ | **0.091** | **0.046** | 0.055 | 0.087 | 0.048 | 0.064 | 0.066 |
| $\Delta F_{ter}$ (Wm$^{-1}$) | 6.38 | 3.73 | 2.89 | **-0.81** | 5.36 | 1.94 | **9.47** |
| Cloud fraction | **0.261** | 0.027 | **0.023** | 0.075 | 0.096 | 0.028 | 0.045 |
| LWP$_{scene}$ (gm$^{-2}$) | **15.34** | 2.19 | 2.44 | 2.24 | 7.39 | **1.92** | 4.32 |
| LWP$_{cloud}$ (gm$^{-2}$) | 69.73 | **106.43** | 90.63 | 77.77 | 76.81 | **62.82** | 67.64 |
| r$_{eff}$ (µm) | **19.68** | 14.06 | **12.02** | 12.22 | 19.03 | 13.26 | 13.18 |
| I$_{org}$ | 0.74 | 0.76 | **0.79** | 0.78 | 0.72 | 0.76 | **0.69** |
| $z_{ct}$ (km) | 1.87 | **1.43** | 1.60 | 1.91 | 2.00 | **2.13** | 1.68 |

and therefore higher amount of liquid water distributed in the scene, which can be attributed to the large "flower" clouds, the clouds themselves had a lower LWP and droplets with a higher r$_{eff}$. This could demonstrate a case wherein the macrophysical

and microphysical properties counteract each other and raises new questions to be answered. For example, does this always happen or are there specific requirements for this? In the case of the 31 January flight, do the microphysical properties make a contribution to $\alpha_{ce}$ at all? If, for example, the r$_{eff}$ had been larger, would the $\alpha_{ce}$ have been noticeably lower?

Another feature of the measurements collected on 7 February that is worth noting in Fig. 4 is that the percentile line for cloud fraction reveals that a relatively large range of cloud fractions were sampled throughout the flight. This is true also for

the flights on 24 January, 2 February and 11 February. The implication here is that a more diverse range of clouds and/or dynamic situations were encountered and sampled in a single flight, unlike 31 January, for example. Therefore, Fig. 4 may be an incomplete picture as the inclusion of multiple cloud situations in this statistical view could certainly make the interpretation more difficult.

The terrestrial component of the cloud radiative effect has a different relationship to the same cloud properties. Fig. 5

contains a series of scatter plots showing the general relationship of different cloud properties to the $\Delta F_{ter}$ calculated using Eq. 1. The majority of the calculated $\Delta F_{ter}$ values are positive, indicating a weak warming effect above the clouds, with some indication of weak cooling for the lowest clouds (below 2.5 km). This weak warming is typical for low-level clouds given the low temperature contrast to the warm surface (Chen et al., 2000). The cooling, however, might fit into the uncertainties of the observations and simulations, as mentioned in Sect. 2.3. Similar to $\alpha_{ce}$, the cloud fraction (Fig. 5(a)) appears to be strongly

tied to the resulting $\Delta F_{ter}$. The relationship to $z_{ct}$ also appears more pronounced, showing that $z_{ct}$ is linked to a decrease in cloud top temperature as expected (Fig. 5(f)). Consequently, the cloud warming effect increases in magnitude. In contrast to what is shown in Fig. 4, both the LWP$_{scene}$ and LWP$_{cloud}$ (Fig. 5(b) and (c)) show a difficult to interpret relationship to $\Delta F_{ter}$. Also, the relation between r$_{eff}$ and I$_{org}$ to $\Delta F_{ter}$ (panels (d) and (e)) is potentially masked by the dominating effects of cloud fraction and $z_{ct}$. Additional methods are required to untangle these relationships.





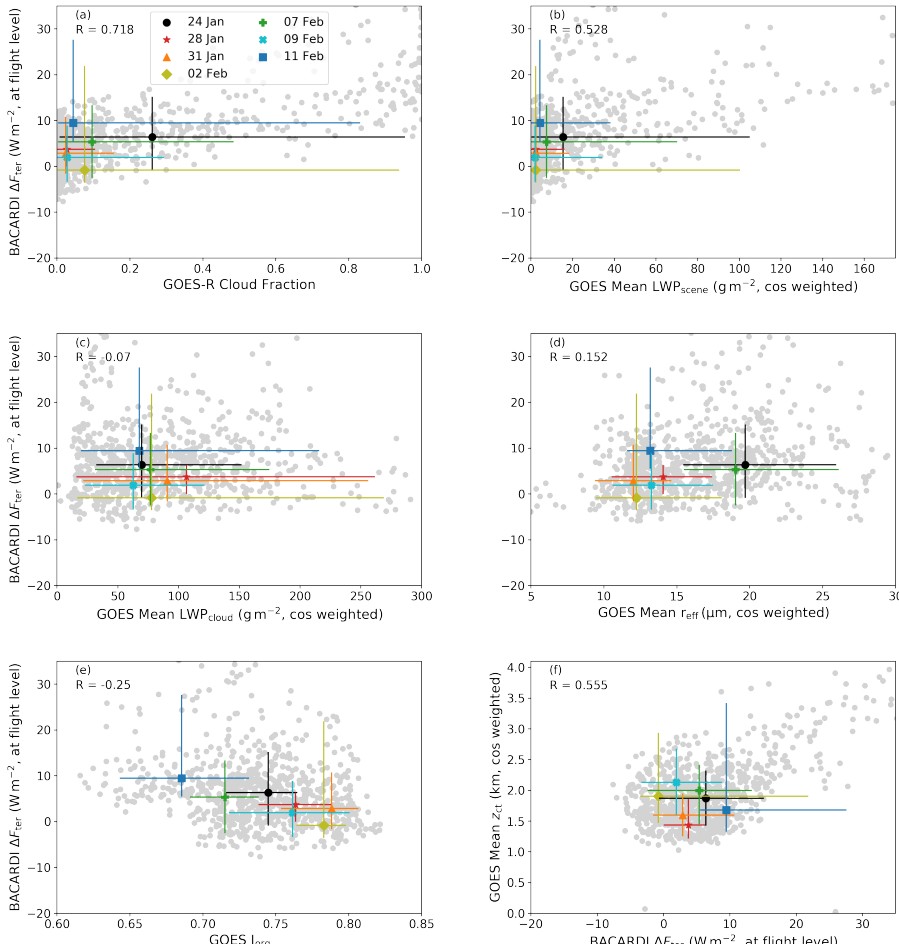

**Figure 5.** Same as Fig. 4 but for $\Delta F_{\text{ter}}$.

In Fig. 5, the lowest median $\Delta F_{\text{ter}}$ was observed during the flight conducted on 2 February (yellow diamond), while the highest was on 11 February (blue square). The order of flights by increasing median $\Delta F_{\text{ter}}$ is different than that of increasing $\alpha_{\text{ce}}$ in Fig. 4. In contrast to $\alpha_{\text{ce}}$, the pattern of increasing median $\Delta F_{\text{ter}}$ with increasing cloud fraction is not as clear. The $z_{\text{ct}}$ relationship, which is more clear in the scatter plot in Fig. 5 is not clearly captured in the pattern of the individual flights. This is perhaps due to the fact that between days, other properties like the atmospheric profile are changing, which could mask the $z_{\text{ct}}$ effect. For a constant temperature and humidity profile between days, we would expect the relation between $\Delta F_{\text{ter}}$ and $z_{\text{ct}}$ to be more clear among the individual flights. Although it is poorly correlated in the subset of data as a whole, the most defined relationship found here between individual flights is with the organization of the cloud field, $I_{\text{org}}$. As the cloud field becomes more clustered, $\Delta F_{\text{ter}}$ decreases. This is in contrast to the results of Tobin et al. (2012), but their results pertain to deep convection, and here shallow clouds are the focus.





Looking again at some flights individually, we can deduce why the observations from different flights may look the way they do. If we look at the flight on 24 January (black circle), this flight stands out in Fig. 5(a) and (b) as having the highest median cloud fraction and $LWP_{scene}$, but not the highest $\Delta F_{ter}$. The $LWP_{cloud}$ is relatively low and the $R_{reff}$ relatively large, and the cloud field has a middle range clustering compared to the other flights. The $z_{ct}$ is also in a middle range. Using cloud fraction alone, we would expect a higher $\Delta F_{ter}$ relative to other clouds, but based on the results in this figure, the $I_{org}$ or $z_{ct}$ could

be factors dampening or overtaking the impact of cloud fraction. The high $z_{ct}$ on the 9 February flight may also be having a similar impact since the median $\Delta F_{ter}$ is high considering the low median cloud fraction and $LWP_{scene}$ also observed during that flight.

### 3.3   Standardized regression analysis

To better understand the relative influence that different properties or the organization of the cloud field have, we use a pa-

rameter known as a beta coefficient (e.g., Neter et al., 1983; Bring, 1994). A beta coefficient represents the slope of the linear regression calculated for variables that have been standardized, so that they have a mean of 0 and a standard deviation of 1. Standardization is simply carried out with

$$Z = \frac{x - \mu}{\sigma}, \tag{7}$$

where $Z$ is the standardized version of variable $x$, $\mu$ is the mean and $\sigma$ is the standard deviation. The benefit of using this

method is that the variables are independent of their units, and the slopes of the linear regression of each cloud parameter with $\alpha_{ce}$ and $\Delta F_{ter}$ can be compared, thus indicating how much a change in any one variable leads to a change in $\alpha_{ce}$ or $\Delta F_{ter}$, respectively.

    Figure 6 shows the beta coefficients for each variable for the subset of flights as a whole as well as for individual flights. Cloud fraction and $LWP_{scene}$ show the highest correlation to $\alpha_{ce}$ for the subset of flights anaylzed here (all flights) and for the

individual flights. $z_{ct}$ is also relatively high for all flights and some individual flights, whereas the correlations of the remaining parameters are low in most cases. It becomes clear that cloud fraction and $LWP_{scene}$ are the main drivers of $\alpha_{ce}$ for both the campaign data set as a whole and for individual flights. The microphysical properties, on the other hand, are less straightforward to interpret. For all flights, neither $LWP_{cloud}$ nor $r_{eff}$ demonstrate a strong correlation to $\alpha_{ce}$. The organization of the cloud field produces a similarly weak correlation.

Looking at individual flights, the overall correlation of $LWP_{cloud}$ is weakly positive, but for flights on the 24 and 28 January the $LWP_{cloud}$ demonstrates a stronger influence on the $\alpha_{ce}$. $r_{eff}$ is also an interesting parameter due to the fact that overall and in some individual flight cases, it has a positive correlation to $\alpha_{ce}$, while in other cases, like 31 January, the correlation is negative. Also, the importance of $z_{ct}$ varies between flights, while the importance of $I_{org}$ maintains a low level of correlation to $\alpha_{ce}$.

    In the case of $\Delta F_{ter}$, for all flights and most of the individual flights, it is clear that the macrophysical properties are also the

most highly correlated to $\Delta F_{ter}$, although not as strongly as with $\alpha_{ce}$. The 28 January and 9 February show the $r_{eff}$ and $I_{org}$ to have the largest impact on $\Delta F_{ter}$, and with a magnitude stronger than what was observed for $\alpha_{ce}$. $z_{ct}$ is also generally strongly positively correlated with $\Delta F_{ter}$. $I_{org}$ often demonstrates a stronger correlation to $\Delta F_{ter}$ than $\alpha_{ce}$, but the positive or negative





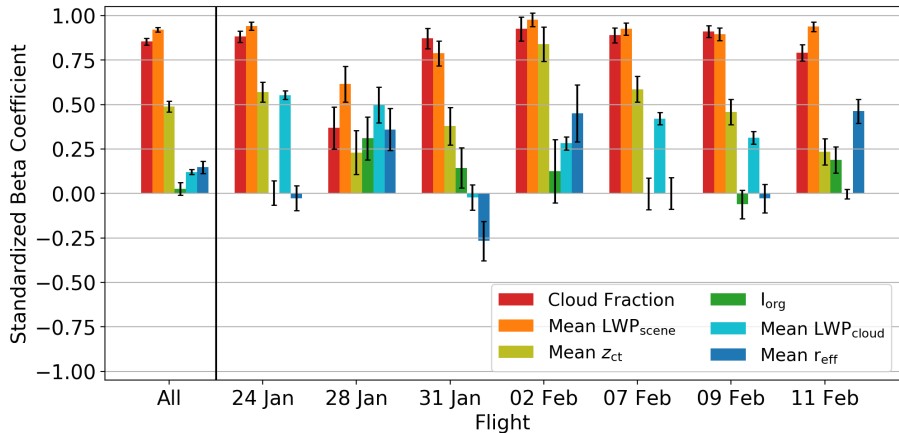

**Figure 6.** Beta coefficients demonstrating the relative correlation of different cloud properties and the organization of the cloud field to $\alpha_{ce}$ for the subset of flights (all) and individual flights.

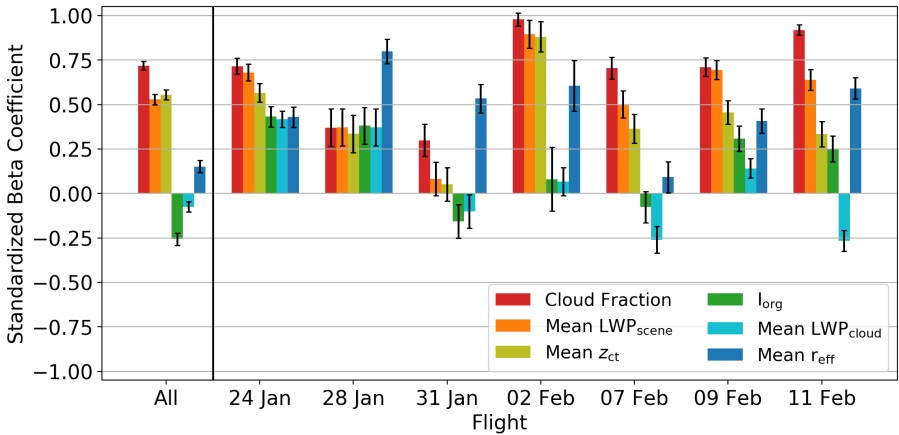

**Figure 7.** Same as Fig. 6 but for $\Delta F_{ter}$.

sign of that correlation changes from flight to flight, which suggests that the effects of this parameter could be linked to other properties or processes. Looking at all flights shows that correlation to be negative. LWP$_{cloud}$ shows a mostly minor correlation

with $\Delta F_{ter}$.

It should be noted that while the beta coefficients are a useful tool for pointing out the correlation of the different variables to $\alpha_{ce}$ and $\Delta F_{ter}$, they are not necessarily good indicators of how the different variables work in concert with each other. Furthermore, the differences between flights rely very strongly on the individual distribution of the data for each flight. Because of the limited amount of data from each flight, outliers can significantly influence the results of a linear regression approach.

In some cases it is not possible or is unreliable as evidenced by the large error bars in Figs. 6 and 7. Thus, such results should be interpreted with caution.





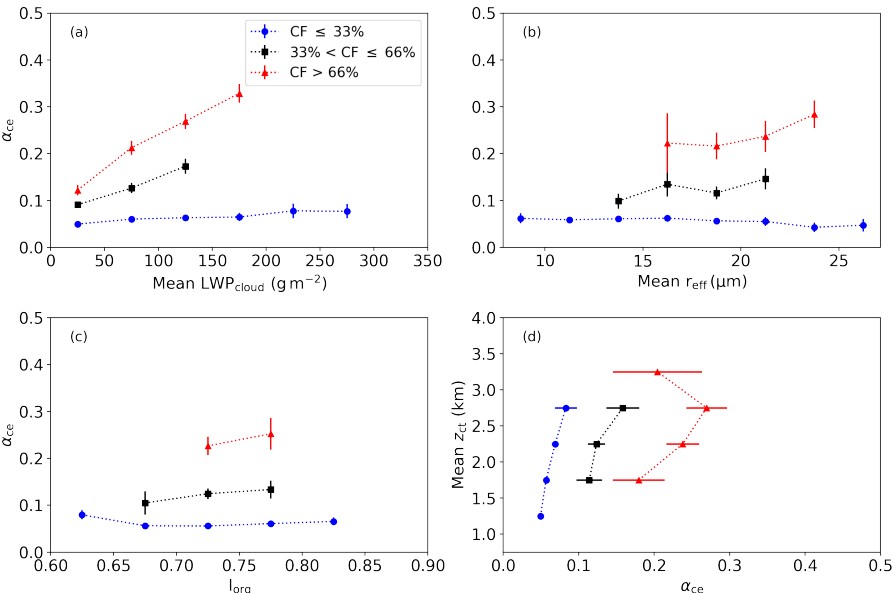

**Figure 8.** The relationship between $\alpha_{ce}$ and different cloud properties also as a function of different cloud fraction classifications. (a) LWP$_{cloud}$, (b) r$_{eff}$, (c) I$_{org}$ and (d) $z_{ct}$. The bars indicate 95% confidence intervals.

### 3.4 Separation into cloud fraction regimes

The analysis has shown that macrophysical properties, cloud fraction and LWP$_{scene}$, are the main drivers of the $\alpha_{ce}$. To extract the impact of the microphysical properties, the data is divided into different cloud fraction classes – low cloud fraction for

values below 33 %, middle cloud fraction for values between 33 % and 66 % and high cloud fraction for values greater than 66 %. This approach also allows for the exploration of those cases shown in Figs. 4 and 5 where the percentile range lines reveal that a large range of cloud fraction values were sampled in a single flight. The data is further categorized into different LWP$_{cloud}$, r$_{eff}$, I$_{org}$ and $z_{ct}$ ranges, and the results are shown in Fig. 8. It should be noted that due to the limited amount of data, this evaluation is not carried out for individual flights, where we might expect to see certain details expressed differently for

different situations.

The range of values observed in the data is of course much more diverse than what is shown here, but only small amounts of data are found at the extremes of those ranges. Thus, in an effort to preserve the statistical representativeness of the data, only cloud property bins with at least 10 data points are shown. This explains why the lowest cloud fraction bin has the largest ranges among the different cloud properties; the majority of the data observed during this campaign had low cloud fractions.

From this figure, the effect that different cloud properties have on cloud scenes of similar cloud fractions can be ascertained. For all cloud properties, an increase or decrease in that property has a limited effect on the $\alpha_{ce}$ when the cloud fractions are low. Specifically, there is a weak increase in $\alpha_{ce}$ with increasing LWP$_{cloud}$ and $z_{ct}$ and a weak decrease in $\alpha_{ce}$ with increasing r$_{eff}$. In the case of LWP$_{cloud}$, this low cloud fraction line coincides with the consistent low $\alpha_{ce}$ mode in Fig. 4(c). The impact





of $I_{org}$ on $\alpha_{ce}$ for low cloud fractions is not clearly defined. When the cloud fractions are in the middle range, the impact of
increasing $LWP_{cloud}$ and $z_{ct}$ becomes stronger, resulting in higher $\alpha_{ce}$. Additionally, both $r_{eff}$ and $I_{org}$ show a weak positive
correlation with $\alpha_{ce}$. This is the first indication from the analyses presented here that the organization of the cloud field, in
terms of the degree of clustering, has an impact on the solar radiative effects of the trade-wind cumuli. For high cloud fraction
cases, the impact of all properties shown in Fig. 8 increases as indicated by the stronger slopes. With respect to $LWP_{cloud}$, this
characterizes the mode in Fig. 4(c), where an increase in $LWP_{cloud}$ coincides with an increase in $\alpha_{ce}$. Notably, around 3 km, the
$z_{ct}$ reverses its correlation to $\alpha_{ce}$ from positive to negative. This may be due to the limited amount of data in this cloud fraction
and $z_{ct}$ combination, which subsequently limits the representation of this group.

This result leads to the conclusion that while macrophysical properties are the main driver of the $\alpha_{ce}$, as cloud fraction
increases, the microphysical properties as well as the horizontal cloud field organization and $z_{ct}$ make a greater contribution to
the resulting $\alpha_{ce}$. When cloud fraction values are low, the impact of those properties, regardless of magnitude, is minor.

Figure 9 shows how the relationship of different cloud properties to $\Delta F_{ter}$ changes when the cloud fraction is held constant.
In the low cloud fraction group, the microphysical properties $LWP_{cloud}$ and $r_{eff}$ appear to have no impact on $\Delta F_{ter}$ regardless
of their magnitude. The impact of increasing $z_{ct}$ is also minimal. The organization of the cloud field shows a clearly negative
impact on $\Delta F_{ter}$ for this cloud fraction class; as the clustering of the cloud field increases, the $\Delta F_{ter}$ decreases. For the middle
cloud fraction range, the negative correlation to $I_{org}$ continues to be present. The relationship of $\Delta F_{ter}$ to $LWP_{cloud}$ and $r_{eff}$
also shows a negative correlation. In this cloud fraction range and the high cloud fraction range, the relationship of $\Delta F_{ter}$ to
$z_{ct}$ is even stronger. The impact of $I_{org}$ continues to have a negative correlation, but does not appear to increase or decrease
in strength relative to the other cloud fraction groups. The impacts of $LWP_{cloud}$ and $r_{eff}$ are unclear for the high cloud fraction
range. Given that they do not decrease with increasing $\Delta F_{ter}$, like in the middle cloud fraction regime, this could suggest that
competing mechanisms are represented here.

## 350 4 Discussion and Conclusions

Bringing together a comprehensive characterization of trade-wind clouds based on their macrophysical and microphysical
properties and interactions is a challenge, one component of which is the lack of representative observational data sets. The
recent EUREC$^4$A campaign provides a solid database for this task. In this study, we assess how different cloud properties
(cloud fraction, $LWP_{scene}$, $LWP_{cloud}$, $r_{eff}$, $z_{ct}$ and $I_{org}$) affect the cloud radiative forcing of shallow trade-wind clouds and to
what degree these parameters matter relative to each other. Using irradiance observations from the BACARDI broadband
radiometer onboard HALO, we calculate the solar and terrestrial cloud radiative effects, here represented by $\alpha_{ce}$ and $\Delta F_{ter}$.
The cloud properties are obtained via 1-minute collocated cloud products from GOES-16.

The relationships between cloud radiative forcing and different cloud properties are complex and a challenge to disentangle.
However, we have demonstrated that it is possible to use a combination of airborne irradiance measurements and satellite-based
cloud property observations for this purpose. For both the solar and terrestrial component, we find that the macrophysical
properties, cloud fraction and $LWP_{scene}$, are the main drivers of changes in the radiative forcing of trade-wind clouds. For the



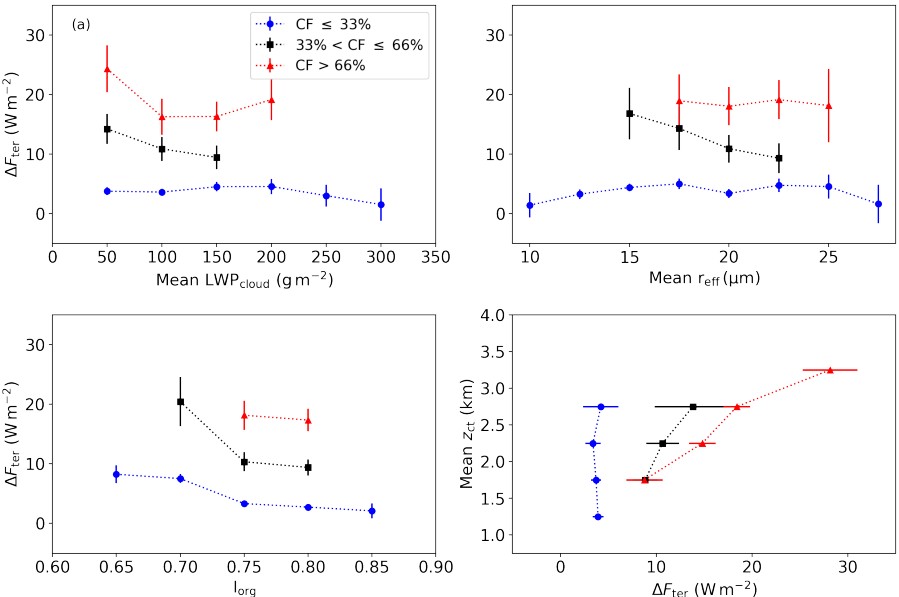

**Figure 9.** Same as Fig. 8 but for $\Delta F_{\text{ter}}$.

solar component, in cases where the cloud fraction is below 33 %, we observe that cloud microphysical properties, $z_{\text{ct}}$ and the organization of the cloud field are of minor importance for determining the cloud radiative forcing, demonstrating weak or no impact on the $\alpha_{\text{ce}}$. As the cloud fraction increases, the relevance of those additional quantities also increases. For the terrestial

component, the impact of $z_{\text{ct}}$ and $\Delta F_{\text{ter}}$ strengthens with increasing cloud fraction. $I_{\text{org}}$ also demonstrates a consistent negative impact on $\Delta F_{\text{ter}}$ at any cloud fraction, while the impacts of $LWP_{\text{cloud}}$ and $r_{\text{eff}}$ are more difficult to interpret.

In comparison to each other, the solar component is more strongly positively correlated to $LWP_{\text{cloud}}$ and $r_{\text{eff}}$. The correlations to $z_{\text{ct}}$ and $I_{\text{org}}$ are also weakly positive for $\alpha_{\text{ce}}$. This is in contrast to $\Delta F_{\text{ter}}$, where the positive correlation to $z_{\text{ct}}$ is much stronger, particularly as cloud fraction increases, and the correlation to $I_{\text{org}}$ is also strong, but negative. This last point indicates that as the

cloud field becomes more clustered, the terrestrial cloud radiative effects decrease, while the $\alpha_{\text{ce}}$ increases slightly. With these points in mind, it should be possible to discern a general idea of the cloud radiative forcing by looking only at the properties of a cloud field based on satellite measurements.

In those instances where certain relations were unclear, it could be possible that the addition of other information or ways of categorizing the data could be the key to extracting more clear results. Here, the results should be constrained to specific

cloud types observed during EUREC$^4$A, which are largely a mixture of shallow cumulus and low stratiform clouds. This could be further extended into categorization based on the description of four cloud types given in Schulz et al. (2021). Also, using $I_{\text{org}}$ as the parameter for defining the organization of the cloud field may do so correctly in a macrophysical sense, but it fails to capture other important qualities about the clouds, such as the optical thickness, which other methods for describing cloud morphology capture more effectively (McCoy et al., 2017). Furthermore, the complexities of better understanding the relation

of $r_{eff}$ to cloud radiative forcing may be found by separating precipitating versus non-precipitating clouds (Lohmann et al., 2000).

Another point to consider is the fact that the cloud fraction, organization of the cloud field, etc. may not be accurately depicted due to the coarse resolution of GOES-16. Coarse resolution leads to missing small clouds, cloud fractions that are overestimated and reflectances that are falsely attributed to cloudy and cloud-free pixels (Koren et al., 2008). However, because

the cloud radiative forcing in this study is calculated from airborne observations, we do not expect the irradiance observations to be falsely representative of cloudy sky conditions. Also, by adjusting the BACARDI footprint used for this study to be compatible with the reflectance observations of GOES, we are ensuring a fair comparison. Nevertheless, the question remains concerning whether higher resolution observations of cloud properties are necessary for determining what cloud properties drive the cloud radiative forcing. To answer this, additional work is planned for a subsequent study including imaging remote

sensing at a high spatial resolution (below 10 m) such as the VELOX thermal IR camera that was also present onboard HALO during EUREC$^4$A.

*Data availability.* Time series of the BACARDI observations are published on AERIS by Ehrlich et al. (2021). All other data produced for this study from BACARDI are available upon request from the leading author. The dropsonde data used in this study are published on AERIS by George (2021). The source for the GOES-16 data set used in this study is the National Oceanic and Atmospheric Administration

(NOAA)/National Environmental Satellite, Data, and Information Service (NESDIS) and the University of Wisconsin-Madison/Cooperative Institute for Meteorological Satellite Studies (CIMSS).

*Author contributions.* AL is the primary author of the paper. AE, MS, KW, and MW carried out the airborne experimental work. Simulations with libRadtran were performed by KW and AE. AL, AE, and KW analyzed and compared the observations and simulations. All authors contributed to the interpretation of results and wrote the paper.

*Competing interests.* The authors declare that they have no conflict of interest.

*Acknowledgements.* This work was supported by the Deutsche Forschungsgemeinschaft (DFG) within the HALO SPP 1294 (project numbers 422897361 and 316646266). We thank the Max Planck Institute for Meteorology for designing and coordinating the EUREC$^4$A campaign and the German Aerospace Center (Deutsches Luft und Raumfahrtzentrum, DLR) for campaign support. We also thank the NOAA/NESDIS and University of Wisconsin-Madison/CIMSS team for providing the dedicated EUREC$^4$A GOES-16 data set. We are grateful to H. Deneke

and M. Klingebiel for helpful discussions and support that contributed to this work.





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
