# Peer review of "An assessment of macrophysical and microphysical cloud properties driving radiative forcing of shallow trade-wind clouds"

_Atmospheric Chemistry and Physics, 2021_

## Referee Comment (RC1)

Comments to the manuscript with ID number "acp-2021-812"

General comments:

The authors used the airborne measurements of cloud radiative-feedback to explore "how and to what degree the macrophysical and microphysical cloud properties influence the cloud radiative effects of shallow trade-wind clouds". By using a combination of the BACARDI measurements and GOES-16 retrievals, they showed that the cloud radiative-feedback is mainly driven by macro-physical properties of these cloud type.

The manuscript is well written and organized with clear scientific questions challenged and robust answers provided as summarized above. The authors also carefully discussed the limitations of their study and opened the door for future investigation.

With this, I would recommend the publication of this manuscript with minor revision.

Specific comments:

0. L50: The authors stated that "… irradiance is not measured directly, which is one reason why airborne observations of this quantity are so important …". Did the authors mean that the airborne can measure the irradiance directly? If so, could you please briefly explain how as this is a highlight of your study compared to other studies only using satellite retrievals?
1. L.89: Why is the circle pattern independent of the meteorological situation?
2. L.92: by BACARDI?
3. L.96: Do the uncertainties depend on the wavelength? Are these instrumental uncertainties? If so, could you please provide references?
4. L.157-167: Fig.1 and the corresponding discussion could be moved to the appendix since it is only a quality check of the BACARDI measurements and GOES-16 retrievals. Would it be more illustrative to provide the exact measurement time (in UTC), location (lat, lon , and the flight height), and the measurement area in the caption of Figure 1?
5. L.349: Could you please spell out the possible competing mechanisms explicitly?

Technical corrections:
1. L.84: in-situ
2. Fig 3: add the y-ticks in the rhs column of the figure to improve the readability.
3. Fig.9: the numbering [(b), (c), (d)] appears to be missing.